# Risk Factors for Hospital Readmission for *Clostridioides difficile* Infection: A Statewide Retrospective Cohort Study

**DOI:** 10.3390/pathogens11050555

**Published:** 2022-05-08

**Authors:** Gregorio Benitez, Fadi Shehadeh, Markos Kalligeros, Evangelia K. Mylona, Quynh-Lam Tran, Ioannis M. Zacharioudakis, Eleftherios Mylonakis

**Affiliations:** 1Infectious Diseases Division, Warren Alpert Medical School of Brown University, Providence, RI 02903, USA; gbenitez@lifespan.org (G.B.); markos_kalligeros@brown.edu (M.K.); emylona1@lifespan.org (E.K.M.); qtran@lifespan.org (Q.-L.T.); 2School of Electrical and Computer Engineering, National Technical University of Athens, 15780 Athens, Greece; 3Division of Infectious Diseases and Immunology, Department of Medicine, New York University School of Medicine, New York, NY 10016, USA; ioannis.zacharioudakis@nyulangone.org

**Keywords:** *Clostridioides difficile*, risk factors, readmission, fluoroquinolone, comorbidity, discharge

## Abstract

(1) Background: *Clostridioides difficile* infection (CDI) is associated with a high recurrence rate, and a significant proportion of patients with CDI are readmitted following discharge. We aimed to identify the risk factors for CDI-related readmission within 90 days following an index hospital stay for CDI. (2) Methods: We analyzed the electronic medical data of admitted patients in our health system over a two-year period. A multivariate logistic regression model, supplemented with bias-corrected and accelerated confidence intervals (BCa-CI), was implemented to assess the risk factors. (3) Results: A total of 1253 adult CDI index cases were included in the analysis. The readmission rate for CDI within 90 days of discharge was 11% (140/1253). The risk factors for CDI-related readmission were fluoroquinolone exposure within 90 days before the day of index CDI diagnosis (aOR: 1.58, 95% CI: 1.05–2.37), higher Elixhauser comorbidity score (aOR: 1.05, 95% CI: 1.02–1.07), and being discharged home (aOR: 1.64, 95% CI: 1.06–2.54). In contrast, a longer length of index stay (aOR: 0.97, 95% BCa-CI: 0.95–0.99) was associated with reduced odds of readmission for CDI. (4) Conclusion: More than 1 out of 10 patients were readmitted for CDI following an index hospital stay for CDI. Patients with recent previous fluoroquinolone exposure, greater overall comorbidity burden, and those discharged home are at higher risk of readmission for CDI.

## 1. Introduction

The clinical severity of *Clostridioides difficile* infection (CDI) ranges from asymptomatic colonization to severe illness that may result in death [1]. The incidence of CDI is associated with antibiotic use [2], while the severity of CDI is associated with advanced age [3]. Although CDI has mainly been considered a nosocomial infection, the incidence of CDI has become more prominent in community settings [4], with almost half of cases in the USA originating in the community [5].

Recurrent CDI occurs when symptoms reappear within 8 weeks following clearance of an initial episode of CDI [6,7]. Recurrence occurs in up to 25% of patients, with the risk of a subsequent recurrence being greater after each recurrence [8,9]. As a result, recurrence contributes to high rates of readmission and associated hospital costs [10,11,12].

In this study, we utilized patient medical data derived from an electronic medical record system of the largest hospital network in the state of Rhode Island, USA, to identify the risk factors associated with readmission for CDI within 90 days of discharge from an index hospital stay for CDI. Due to the recurrent nature of illness and high readmission rate, it is imperative to identify high-risk patients for CDI-related readmission.

## 2. Materials and Methods

### 2.1. Patient Selection

Patients admitted to Rhode Island Hospital, the Miriam Hospital, and Newport Hospital in Rhode Island between 1 January 2016 and 1 August 2018, who were at least 18 years of age at the time of admission, and had a positive *C. difficile* polymerase chain reaction (PCR) test result during an index stay were considered eligible for inclusion. Patients who died during the index stay or were discharged to a hospice were excluded. Patients who received fidaxomicin or did not receive inpatient treatment for CDI were also excluded from analysis.

### 2.2. CDI Index Case Definition

In accordance with the Infectious Disease Society of America (IDSA) and Society for Healthcare Epidemiology of America (SHEA) guidelines [7], our institutional policy allows for the use of a standalone nucleic acid amplification test such as a PCR test for CDI diagnosis if patients have at least three unexplained (i.e., not due to laxative use) and unformed stools in 24 h. To ensure that positive PCR test results were due to active infection, we reviewed the medical charts to confirm that patients had diarrhea and that PCR tests were ordered by the patient care teams for the detection of *C. difficile*. Moreover, to verify that an index stay was not a case of CDI-related readmission, we looked back 90 days from the date of CDI diagnosis to check for previous CDI-related admissions.

### 2.3. Study Variables

We collected data on patient demographics such as age, race/ethnicity, and sex; index hospital stay characteristics such as length of stay, discharge disposition, and inpatient treatment regimen for CDI; patient comorbidity burden quantified by the van Walraven weighted Elixhauser Comorbidity Index score [13,14]; and prescription history of high-risk antibiotics associated with an increased risk of developing CDI.

For our analysis, age was categorized into four groups: 18–44, 45–64, 65–79, and over 80 years of age. Discharge disposition was contingent on patients being discharged home, such as home with or without home health services, or a healthcare facility, such as a long-term care facility (LTCF) or a skilled nursing facility. We also reviewed records for the prescription of high-risk antibiotics including cephalosporins, clindamycin, fluoroquinolones, penicillin, and combinations of penicillin with beta lactamase inhibitors [15,16]. A patient was considered exposed to a high-risk antibiotic if they were prescribed a high-risk antibiotic within 90 days prior to the day of index CDI diagnosis.

During the study period, vancomycin was the preferred treatment regimen for a first episode of CDI [7]. Following a medical chart review, patients were categorized as receiving either: (a) oral vancomycin only, (b) oral or intravenous (IV) metronidazole only, (c) sequential dual therapy if they received oral vancomycin and either oral/IV metronidazole with less than 24 h of overlap, or (d) concurrent dual therapy if they received oral vancomycin and either oral/IV metronidazole concurrently for 24 h or more.

In line with the Centers for Disease Control and Prevention (CDC) guidelines [17], CDI index cases were categorized into three different groups (Table A1). Community-onset healthcare facility-associated and hospital-onset cases were grouped together and further classified as healthcare associated [18].

### 2.4. Study Outcome

All adult patients discharged, except those discharged to a hospice, from an index hospital stay for CDI were considered at risk of readmission. Readmission was defined as an inpatient admission within 90 days of discharge, with either a primary encounter diagnosis of CDI or a positive *C. difficile* PCR test result on Day 1, 2, or 3 of admission. For patients who had multiple readmissions, only the first cases of readmission were included in the analysis.

During the study period, a 10-day antibiotic course was the mainstay therapy for the treatment of CDI [7]. Additionally, recurrent CDI may occur up to 8 weeks following the clearance of an initial episode of CDI. Thus, we selected a 90-day readmission window to maximize the capture of patients who required hospital readmission for CDI following discharge.

### 2.5. Statistical Analysis

The continuous variables were represented as medians with IQRs. Univariate tests of association between patient readmission status and patient/index hospital stay characteristics were performed using Wilcoxon rank sum tests for continuous variables and Pearson’s Chi-square test for categorical variables.

To identify the risk factors for CDI-related readmission, age, sex, race/ethnicity, CDI index case classification, length of index hospital stay, discharge disposition, Elixhauser comorbidity score, inpatient CDI treatment regimen, and exposure to individual high-risk antibiotics prior to index CDI diagnosis were included in a multivariate logistic regression model. The model estimated the adjusted odds ratios (aOR), along with 95% normal-based confidence intervals (95% CI). Bootstrapping of 1000 iterations was then performed to present bias-corrected and accelerated confidence intervals (BCa-CI). Bootstrapping is a random sampling with replacement procedure that uses the original sample to construct a bootstrap distribution that estimates the shape of the sampling distribution of a point estimate [19], such as an odds ratio [20]. BCa-CIs correct for potential bias and skewness of bootstrap distributions and offer more robust coverage than normal-based confidence intervals, since bootstrapping utilizes simulations to circumvent the assumption that the underlying distribution of the point estimate is normally distributed [21,22]. Additionally, we performed a secondary analysis to identify CDI-related readmission risk factors in which readmitted patients without a documented positive *C. difficile* PCR test result during readmission stay were excluded from the analysis. Statistical analyses were performed using Stata/SE 17 (StataCorp, College Station, TX, USA). Statistical significance was defined as *p* < 0.05.

## 3. Results

Between 1 January 2016 and 1 August 2018, a total of 1487 adult CDI cases were documented. A total of 1253 patients were included in the study after we excluded patients as described in Figure 1.

As shown in Table 1, 753/1253 (60%) CDI index cases were classified as healthcare associated, based on the criteria detailed in the Methods section. Moreover, 1020/1253 (81%) patients were Non-Hispanic White, 744/1253 (59%) were discharged home, and 800/1253 (64%) were prescribed a high-risk antibiotic prior to CDI diagnosis. The median length of index stay for all patients was 7 days (IQR: 4–13).

Overall, 140/1253 (11%) patients were readmitted for CDI within 90 days of discharge. Moreover, 114/140 (81%) patients were readmitted with a positive *C. difficile* PCR test result, and 26/140 (19%) were readmitted with a primary diagnosis of CDI in the absence of a PCR test. Following secondary analysis in which readmitted patients without a documented positive *C. difficile* PCR test result during readmission stay were excluded (n = 26), a total of 114/1253 (9%) patients were readmitted within 90 days of discharge (Appendix A).

In our multivariate logistic regression model (Table 2), fluoroquinolone exposure within 90 days prior to the day of index CDI diagnosis, compared with no fluoroquinolone exposure, was associated with an increased risk of readmission for CDI by 58% (aOR: 1.58, 95% CI: 1.05–2.37). No other individual high-risk antibiotic was associated with readmission for CDI. A one-unit increase in the Elixhauser comorbidity score was associated with an increased risk of readmission for CDI of 5% (aOR: 1.05, 95% CI: 1.02–1.07). Furthermore, being discharged home, compared with being discharged to a healthcare facility, was associated with an increased risk of readmission for CDI of 64% (aOR: 1.64, 95% CI: 1.06–2.54).

After both bootstrapping and secondary analysis in which patients without a documented positive *C. difficile* PCR test result during readmission stay were excluded, fluoroquinolone exposure, higher Elixhauser comorbidity score, and being discharged home remained statistically significant risk factors for CDI-related readmission (Table 2 and Appendix A, respectively). Additionally, after bootstrapping, length of index hospital stay was associated with reduced odds of readmission for CDI. Specifically, the odds of readmission for CDI decreased by 3% (aOR: 0.97, 95% BCa-CI: 0.95–0.99) for each additional day spent in the hospital during the index stay.

## 4. Discussion

Utilizing the largest hospital network in the state of Rhode Island, USA, we identified the risk factors for readmission for CDI following an index hospital stay for CDI. More than 1 out of 10 (11%) patients were readmitted for CDI within 90 days of discharge. From the analysis of this cohort, we found that fluoroquinolone exposure within 90 days prior to the day of index CDI diagnosis, higher Elixhauser comorbidity score, and being discharged home were independent risk factors for readmission for CDI. Notably, these three identified risk factors remained statistically significant after excluding patients who were readmitted without a documented positive *C. difficile* PCR test result.

Our rate of readmission (11%) is comparable to the rate reported by Psoinos et al., who analyzed a national sample of Medicare beneficiaries and found that 13% of patients were readmitted for CDI within 90 days of discharge from an index stay for CDI [23]. Moreover, our finding that fluoroquinolone exposure prior to index CDI diagnosis was a risk factor for CDI-related readmission extends the established connection between fluoroquinolone use and CDI. Additionally, our finding that being discharged home was a risk factor for CDI-related readmission is consistent with previous studies [24,25,26]. Lastly, a greater number of Elixhauser-related comorbidities [23,25] and individual comorbidities such as renal failure [24,25,27] and inflammatory bowel disease [10] are risk factors for CDI-related readmission. We found that a higher Elixhauser comorbidity score, as a marker for overall patient comorbidity burden, was a risk factor for CDI-related readmission.

Appropriate antibiotic use is critical for the proper treatment [28] and effective chemoprophylaxis [29,30] of bacterial infections. Institution-based interventions [31] and clinical patient parameters [32] can guide the proper prescription of antibiotics. During periods of severe healthcare strain, such as the ongoing novel coronavirus disease 2019 (COVID-19) pandemic, the potential consequences of misusing antibiotics are notable [33]. The significance of continued appropriate antibiotic use through antimicrobial stewardship programs is highlighted in its association with the reduced incidence of CDI [34]. In particular, the relationship between CDI and fluoroquinolones is well established, with fluoroquinolones associated with a high risk of developing CDI [15,16] and CDI recurrence [35]. A reduction in fluoroquinolone prescriptions has contributed to decreased CDI incidence in England [36] and the USA [37]. In a previous study, members of our team found that a decrease in the rate of antibiotic prescription, particularly fluoroquinolones, was associated with a decrease in the incidence of hospital-onset CDI [38]. However, if a fluoroquinolone must be prescribed, then its selection based on appropriate antimicrobial stewardship programs might minimize the risk of patients developing CDI [39]. Our study finding that fluoroquinolone exposure is a risk factor for CDI-related readmission strengthens the argument for judicious prescription of fluoroquinolones and the need for close monitoring by antimicrobial stewardship.

We found that a unit increase in the Elixhauser comorbidity score was associated with an increased risk of readmission. Comorbidity burden is associated with developing CDI [40,41] and readmission for CDI, with the risk of readmission greatest among patients with at least three Elixhauser-related comorbidities [23,25]. For this study, we opted to use a weighted Elixhauser comorbidity score that quantified the overall patient comorbidity burden [13]. Thus, our finding that a higher Elixhauser comorbidity score is a risk factor for CDI-related readmission adds utility to the established connection between comorbidities and CDI by expanding on previous studies that evaluated either individual [24] or number of comorbidities [23,25] without quantifying the burden. Our method of quantifying the comorbidity burden avoids analyzing multiple individual comorbidities, which increases the risk of overfitting the regression model, and categorizing patients based on the number of Elixhauser comorbidities, which assumes that each Elixhauser comorbidity affects the outcome equally.

Compared with community settings, healthcare settings such as LTCFs are linked with a higher prevalence of *C. difficile* colonization among residents [42] and increased incidence of both initial and recurrent CDI [43,44,45]. However, we found that patients discharged home are more likely to be readmitted for CDI. To explain this finding, healthcare facilities may be better equipped to follow post-discharge instructions, such as ensuring treatment adherence, and manage CDI-related symptoms before readmission is warranted [26]. Similar supportive and comprehensive follow-up monitoring should be accessible to patients who are discharged home. Costantino et al. conducted an intervention in which outpatient calls were made to patients discharged home after any-cause hospitalization [46]. Patients were asked whether they understood which signs/symptoms to be alert to, filled their outpatient prescriptions, and scheduled the necessary follow-up visits. The researchers found that the outpatient calls, particularly if conducted closer to the time of discharge, reduced the rate of all-cause 30-day readmissions [46]. Given that patients who are discharged home are at greater risk of CDI-related readmission, similar close-monitoring outpatient interventions to monitor treatment adherence and manage disease progression should be evaluated and implemented if they are effective in minimizing the rate of CDI-related readmissions.

Interestingly, we found that in our cohort, a longer length of index stay was associated with reduced odds of readmission for CDI. Thus, an extended index stay may serve as a protective factor, instead of a risk factor, for CDI-related readmission. However, CDI-related hospital stays are associated with greater financial cost [47,48], so prolonging hospital stays for patients is not practical. Therefore, the need for effective outpatient interventions that minimize the rate of CDI-related readmissions is further emphasized.

Our study offers a novel evaluation of potential CDI-related readmission risk factors not available in previous studies that evaluated the risk factors for CDI-related readmission following index stay for CDI [24,25,27]. For instance, our model accounts for the inpatient treatment regimen for CDI to assess whether the specific treatment regimen that a patient received is associated with readmission, the timing of *C. difficile* specimen collection to assess whether patients with community- and healthcare-associated CDI are differentially at risk for readmission, and high-risk antibiotic use prior to index CDI diagnosis to assess whether exposure to a high-risk antibiotic is associated with readmission. Moreover, a commonly reported study limitation of previous studies [23,24,25,27] that relied on administrative claim data is that using the International Classification of Disease codes to identify both CDI index and readmission cases may underestimate the number of CDI cases, resulting in missed cases. Access to patient medical charts allowed us to identify patients who had a positive *C. difficile* PCR test result during their index stay to maximize the number of CDI cases for analysis. For readmission cases, we relied on positive PCR test results for identification, along with primary encounter diagnoses if repeat testing was not conducted during readmission.

Since laboratory testing cannot differentiate between *C. difficile* colonization and active infection [49], patients may have a positive PCR test result in the absence of active infection. To confirm that a patient was an index CDI case, we performed a medical chart review to verify that patients with a positive *C. difficile* PCR test result during their index stay had diarrhea, which is in accordance with IDSA and SHEA guidelines [7], as described in the Methods section. However, while laboratory confirmation is recommended before treating patients for suspected recurrence [7], a positive PCR test result alone without testing for toxin production among patients readmitted with diarrhea may result in an overestimation of readmission cases.

Regarding study limitations, causality cannot be inferred from the associations described due to the retrospective nature of the study, with the objective of identifying the risk factors and not the causal mechanisms of action. In addition, we did not have access to outside records, so patients may have had a positive *C. difficile* PCR test result prior to readmission. Thus, we reviewed the medical chart data of all index cases readmitted within 90 days of discharge to identify readmission cases who had a principal encounter diagnosis of CDI without a documented PCR test. However, relying on primary encounter diagnoses, which are based on clinical judgement after considering patients’ symptoms and CDI history, may also result in overestimation of readmission cases, since we do not have laboratory confirmation that the causative agent of diarrhea was *C. difficile.* In addition, laboratory values such as serum creatinine level and leukocyte count that are markers of CDI severity [7] were not included in our model, since they were not consistently reported for patients. We also did not analyze outpatient use of proton pump inhibitors or high-risk antibiotic use during or following discharge from the index stay for CDI. Furthermore, our high-risk antibiotic data were limited to a binary categorization of either having been prescribed or not prescribed individual high-risk antibiotics during the specified period. The claim data did not provide reliable data on the number of days a patient was prescribed high-risk antibiotics, and we could not verify whether patients followed prescription orders. Lastly, our study results may have limited generalizability, since the cohort only consisted of patients from hospitals in our state.

## 5. Conclusions

This study identified three main risk factors for readmission for CDI: fluoroquinolone exposure within 90 days prior to the day of index CDI diagnosis, higher Elixhauser comorbidity score as a measure of overall patient comorbidity burden, and being discharged home following an index stay. Identifying high-risk patients for readmission for CDI is imperative due to the significant rate of costly CDI-related readmissions. Continued antimicrobial stewardship efforts such as close monitoring of fluoroquinolone prescription and the use of alternative agents are critical to sustaining decreased incidence of CDI, and may aid in diminishing the threat of CDI across all levels ranging from initial CDI, recurrent CDI, and inpatient readmission. Additionally, more research is needed to evaluate and implement effective outpatient interventions that minimize the rate of CDI-related readmissions, especially among high-risk patients such as those discharged home and patients with a greater overall comorbidity burden who may benefit from closer monitoring following discharge.

## Figures and Tables

**Figure 1 pathogens-11-00555-f001:**
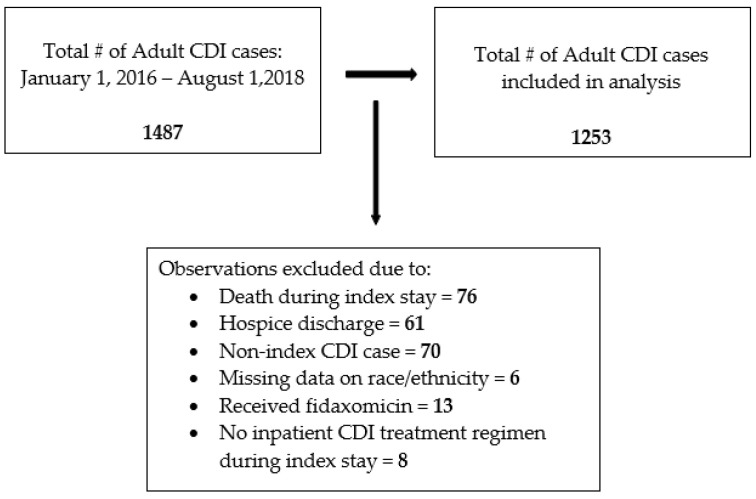
Flow diagram for inclusion of CDI index cases for analysis.

**Table 1 pathogens-11-00555-t001:** Characteristics of CDI index cases.

	Index Cases N (%)	Readmitted for CDI N (%)	Not Readmitted for CDI N (%)	*p*-Value
Overall	1253	140	1113	
**Age**				0.474
18–44	186 (15)	24 (17)	162 (15)	
45–64	373 (30)	46 (33)	327 (29)	
65–79	389 (31)	36 (26)	353 (32)	
Over 80	305 (24)	34 (24)	271 (24)	
**Sex**				0.687
Female	718 (57)	78 (56)	640 (58)	
Male	535 (43)	62 (44)	473 (42)	
**Race/Ethnicity**				0.430
Non-Hispanic White	1020 (81)	107 (76)	913 (82)	
Hispanic or Latino	117 (9)	17 (12)	100 (9)	
Non-Hispanic Black	84 (7)	11 (8)	73 (6)	
Other	32 (3)	5 (4)	27 (3)	
**CDI Index Case** **Classification**				0.479
Community Associated	500 (40)	52 (37)	448 (40)	
Healthcare Associated	753 (60)	88 (63)	665 (60)	
**CDI Treatment Regimen**				0.479
Vancomycin	507 (40)	61 (44)	446 (40)	
Metronidazole	315 (25)	29 (21)	286 (26)	
Sequential	323 (26)	40 (29)	283 (25)	
Concurrent	108 (9)	10 (7)	98 (9)	
**Discharge Disposition**				0.019
Healthcare Facilities	509 (41)	44 (31)	465 (42)	
Home	744 (59)	96 (69)	648 (58)	
**High-Risk Antibiotic °**				0.763
No	453 (36)	49 (35)	404 (36)	
Yes	800 (64)	91 (65)	709 (64)	
**Elixhauser Score**				0.004
Median [IQR]	5 [0–11]	6 [0–13]	5 [0–10]	
**Length of Index Stay (days)**				0.094
Median [IQR]	7 [4–13]	6 [4–10.5]	7 [4–13]	

Abbreviations: IQR: interquartile range, °: prescription of cephalosporins, clindamycin, fluoroquinolones, penicillin, or combination of penicillin with beta lactamase inhibitors within 90 days prior to the day of index CDI diagnosis.

**Table 2 pathogens-11-00555-t002:** Adjusted associations of factors with CDI-related readmission.

	Adjusted Odds Ratio	95% CI	95% BCa-CI
**Age**			
18–44	Reference		
45–64	0.91	0.52–1.58	0.52–1.58
65–79	0.63	0.35–1.15	0.35–1.22
Over 80	0.84	0.45–1.57	0.43–1.65
**Sex**			
Female	Reference		
Male	1.02	0.71–1.47	0.69–1.52
**Race/Ethnicity**			
Non-Hispanic White	Reference		
Hispanic or Latino	1.17	0.65–2.10	0.62–2.04
Non-Hispanic Black	1.15	0.57–2.31	0.54–2.17
Other	1.74	0.64–4.76	0.49– 5.24
**CDI Index Case** **Classification**			
Community Associated	Reference		
Healthcare Associated	1.46	0.95–2.22	0.91–2.29
**CDI Treatment Regimen**			
Vancomycin	Reference		
Metronidazole	0.70	0.42–1.18	0.42–1.13
Sequential	1.13	0.72–1.78	0.76–1.83
Concurrent	0.74	0.35–1.56	0.31–1.38
**Discharge Disposition**			
Healthcare Facilities	Reference		
Home	1.64	1.06–2.54	1.03–2.71
**Fluoroquinolones**			
No	Reference		
Yes	1.58	1.05–2.37	1.01–2.31
**1st/2nd Cephalosporins**			
No	Reference		
Yes	0.82	0.49–1.38	0.47–1.39
**3rd/4th/5th** **Cephalosporins**			
No	Reference		
Yes	0.98	0.65–1.48	0.63–1.48
**Clindamycin**			
No	Reference		
Yes	0.79	0.32–1.95	0.26–1.96
**Penicillin**			
No	Reference		
Yes	0.94	0.46–1.89	0.44–1.95
**Penicillin with Beta Lactamase Inhibitors**			
No	Reference		
Yes	0.99	0.66–1.48	0.67–1.49
**Elixhauser Score**			
Unit Increase	1.05	1.02–1.07	1.01–1.07
**Length of Index Stay (Day)**			
Unit Increase	0.97	0.95–1.00	0.95–0.99

## Data Availability

The datasets generated and/or analyzed during the current study are not publicly available due to HIPAA restrictions. De-identified summary data are available from the corresponding author on reasonable request.

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
