# Peer review of "Risk Factors for Hospital Readmission for Clostridioides difficile Infection: A Statewide Retrospective Cohort Study"

_pathogens, 2022, doi:10.3390/pathogens11050555_

Round 1

Reviewer 1 Report

This study investigated the risk factors for hospital readmission for Clostridioides difficile infection (CDI). This study is well designed, and the results obtained from this study are reliable.

However, similar papers have been published, and thus this manuscript is poor in novelty. If the authors want this manuscript published in this journal, they should present the differences between this manuscript and the previous similar papers.

This study clarified several risk factors for hospital readmission for CDI, based on which, the authors should present the specific strategies to prevent hospital readmission for CDI.

Reviewer 2 Report

This is an intersting paper on a common problem. The aims are clear. The text is well written and easy to read. I have only a few comments, most of them are minor.

MAJOR

You excluded patients who died so it is not logical to use the Elixhauser comorbidity index which refers to in hospital mortality. Why you did noz choose a more general one, e.g. Charlson comorbidity index?

Case definitions are not clear to me:

Readmission is might be overestimated as positive PCR test result is not identical with C.diff infection even in the presence of diarrhoea. Why did not they tested toxin production?

line 133: readmitted with primary diagnosis of CDI in the absence of PCR test? How do you know than the causative agent was C diff.?

I suggest to analyse all the different antibiotics separately, and not group them to non-vancomycin.

No combination therapy was given to any of the patients?

Have you considered sensitivity analysis?

Minor:

line 30: In the first sentence it is strange to list the disease itself (severe ilness) together with 2 risk factors. Maybe it would be more clear if it were separated into two sentence

line 73: ….within 90 days before the …..I think more clear to say: within 90 days after the index CDI  discharge

line 114-116: repetition of the flow chart

line 117: index (CDI diagnosis)

You used IDSA and SHEA guidelines (see line 51-52, while for CDI categorisation the CDC one (line 78-80). Why?

line 127: what you mean by this „DEsignated if…..were prominent in treatment”?

Please explain why patient were treated with antibiotics within 90 days of index discharge?

line 162: what you mean by unique? Several other studies described the role of fluoroquinolones

Round 2

Reviewer 1 Report

I have reviewed the revised version.

The manuscript has been significantly improved.

I recommend that the manuscript be accepted for publication in Pathogens.